# Structural Polymorphism of Guanine Quadruplex-Containing Regions in Human Promoters

**DOI:** 10.3390/ijms232416020

**Published:** 2022-12-16

**Authors:** Christopher Hennecker, Lynn Yamout, Chuyang Zhang, Chenzhi Zhao, David Hiraki, Nicolas Moitessier, Anthony Mittermaier

**Affiliations:** Department of Chemistry, McGill University, Montreal, QC H3A 0B8, Canada

**Keywords:** G4, G4CR, bioinformatics, transcription start site, TSS

## Abstract

Intramolecular guanine quadruplexes (G4s) are non-canonical nucleic acid structures formed by four guanine (G)-rich tracts that assemble into a core of stacked planar tetrads. G4-forming DNA sequences are enriched in gene promoters and are implicated in the control of gene expression. Most G4-forming DNA contains more G residues than can simultaneously be incorporated into the core resulting in a variety of different possible G4 structures. Although this kind of structural polymorphism is well recognized in the literature, there remain unanswered questions regarding possible connections between G4 polymorphism and biological function. Here we report a detailed bioinformatic survey of G4 polymorphism in human gene promoter regions. Our analysis is based on identifying G4-containing regions (G4CRs), which we define as stretches of DNA in which every residue can form part of a G4. We found that G4CRs with higher degrees of polymorphism are more tightly clustered near transcription sites and tend to contain G4s with shorter loops and bulges. Furthermore, we found that G4CRs with well-characterized biological functions tended to be longer and more polymorphic than genome-wide averages. These results represent new evidence linking G4 polymorphism to biological function and provide new criteria for identifying biologically relevant G4-forming regions from genomic data.

## 1. Introduction

Intramolecular guanine quadruplexes (G4s) are four-stranded nucleic acid structures formed when four tracts of contiguous guanine residues, separated by three loops, come together in stacked planar tetrads stabilized by Hoogsteen hydrogen bonding and metal coordination (Figure 1). Putative G4s are plentiful in the human genome and are found in functional regions including origins of replication, introns, 5′ and 3′ untranslated regions, as well as in promoter regions, where they help to regulate gene expression [1,2,3,4,5,6]. The stability of these structures is influenced by a variety of different factors such as the presence of different cations, pH, and molecular crowding [7]. Some of the best-characterized G4s have relatively simple structures consisting of four tracts of three Gs, with all 12 G residues engaged in the core structure [8,9]. However, in general, G4-forming DNA sequences are polymorphic. They contain more Gs than can be simultaneously incorporated into a single structure, resulting in ensembles of different conformations with different subsets of Gs engaged in the core [10,11,12,13]. These polymorphisms can take many different forms. For example, stable G4s can form from G-tracts that contain non-G residues which are bulged out from the core structure [14,15,16]. Alternatively, when the tracts contain different numbers of Gs, the strands can effectively slide with respect to one another, a type of motion we refer to as G-register exchange [17,18,19]. For example, the Pu18 sequence from the human *MYC* promoter (AGGGTGGGGAGGGTGGGG) has two tracts of three Gs and two tracts of four Gs and can form four different G-register isomers: AGGGTGGGGAGGGTGGGG, AGGGTGGGGAGGGTGGGG, AGGGTGGGGAGGGTGGGG, AND AGGGTGGGGAGGGTGGGG, where guanine residues participating in the tetrad core are capitalized and loop residues are in bold type. Of these, the first isomer is the most thermodynamically stable [17]. As well, there are several examples of biologically relevant G4s that contain extra (>4), or “spare tire” G-tracts, such that stable G4s can form from different subsets of four of the tracts [20,21]. For example, the Pu27 sequence from the human *MYC* promoter comprises the Pu18 sequence (above) and a 5th G-tract appended at the 5′ end. Three alternative G4s have been reported for this region in which the tetrad core is formed from tracts 1234, 1245, and 2345 (the Pu18 sequence) [20,22]. Similarly, G-rich minisatellite DNA can contain dozens of consecutive G-tracts [23,24] and can form enormous numbers of different G4 folded states involving different subsets of G-tracts, potentially leading to a highly frustrated energy landscape [25]. Finally, many different chain topologies are possible, with the strands running parallel or antiparallel to one another [26] or adopting snap-back conformations [27]. G4s composed of G-tracts containing between two [28] and six [29] consecutive Gs have been reported. G4s can be stabilized by DNA hairpin formation in the loops and bulges [30,31], additional hydrogen bonding of core guanines to loop residues [32], and stacking of adjacent G4s [33] leading to some highly non-canonical G4 structures [34].

There are several ways in which the polymorphism of G4s has been proposed to impact their biological function [35,36,37,38,39]. For example, we have found that the existence of multiple G register isomers can accelerate folding [19] and stabilize the folded state [17], with implications for G4 function. It has been shown that the presence of spare tire G-tracts can provide resilience to DNA damage [21]. G4-binding proteins such as nucleolin can differentiate between different folding isomers (G4s), whereas different DNA helicases have strong preferences for unwinding different G4 structures [40]. The fact that a single G-rich DNA sequence from a gene promoter region can fold into multiple different structures has been identified as a hurdle to designing effective specific G4-targeting drugs [36]. Many best-characterized G4s with validated biological functions are capable of undergoing both spare tire (>four G-tacts) and G-register (G-tracts of different lengths) isomerism. For example, the well-studied Pu-27 sequence from the promoter of the oncogene *MYC* contains two tracts of GGG and three tracts of GGGG. When eight or more G tracts are present, then in principle two or more adjacent G4s can form. For example, a stretch of 68 nucleotides from the telomerase (*TERT*) gene promoter contains five tracts of GGG and seven tracts of GGGG and can fold into two [41] or three [42] consecutive G4s.

Although seemingly common in naturally occurring G4s, these types of structural polymorphism have not yet been surveyed in a systematic manner, thus their prevalence and functional significance remain poorly understood. Recent reviews have split the bioinformatic approaches for predicting the locations of G4s into the categories of regular expression matching, scoring, sliding window scoring, machine learning, and specialized tools [43]. For example, the quadparser algorithm focuses on identifying short segments of DNA that are likely able to adopt a stable fold by matching the sequence of interest to a consensus motif [44]. Alternatively, the G4Hunter algorithm predicts the tendency of DNA to form G4s by evaluating G richness and G skewness (i.e., the density of tracts containing consecutive Gs) [45]. Newer approaches such as the QPARSE algorithm have identified the prevalence of multimeric G4s in the human genome along with sequences that can form hairpin loops [46]. However, none of these methods quantify the extent of G4 structural polymorphism. Thus, the extent, prevalence, distribution, and functional significance of G4 polymorphism remain largely unknown.

We have set out to evaluate G4 structural polymorphism in promoter regions of the human genome. To begin, we recognized that the basic functional unit is not a single G4 structure, but rather a G-rich stretch of DNA that can adopt between one and a multitude of different G4 folds. We defined a G4-containing region (G4CR) as a contiguous stretch of DNA-containing G-tracts, such that each G-tract can, in principle, form a stable G4 with the G-tracts on either side. In other words, G4CRs may be thought of as regions of DNA where every single residue has the potential to be included in at least one G4, in either a core, bulge, or loop position. We found that G4CRs can be anything from about 15 nucleotides (nt) ((GGGN)_3_GGG) to several hundred nt in length. They can form between one and several thousand structural distinct G4s, with up to about 25 simultaneously adjacent. In parallel, we also calculated the structural multiplicity of individual G residues, which we defined as the number of structural distinct G4s that incorporate a particular G residue into the core. Since multiplicity is defined on a per-G basis, this provides a simple approach for mapping structural polymorphism onto a DNA sequence with single-residue precision. We found that multiplicity values can vary between about one and a thousand for different G residues within a single G4CR. Intriguingly, the degree of polymorphism within a G4CR appears related to both the location of the G4CR relative to the transcription start site, and also the lengths of loops and presence of bulges in the G4 structures it forms. Furthermore, a variety of well-characterized biologically relevant G4-containing regions from the *MYC*, *VEGFA*, *BCL2*, *KIT*, and *KRAS* gene promoters have greater than average degrees of polymorphism. This is new evidence that polymorphism itself may be related to the biological function of G4CRs and provides new criteria for identifying potentially important regulatory sites in DNA.

## 2. Results

### 2.1. Calculating G4 Polymorphism

Predicting the number of possible different G4 conformations available to a single stretch of G-rich DNA becomes intractable if every source of structural variation is considered. For example, we are not aware of any method for reliably predicting *a priori* which topology or topologies (parallel, antiparallel, hybrid) a particular DNA sequence will favor. Non-canonical inter-G4 stacking and additional base-pairing between core and loop and flanking residues are usually only identified by biophysical analyses and full three-dimensional structural elucidation [47], although this is complicated somewhat by the fact that crystal packing can influence G4 structure [48]. Therefore, we have opted to simplify the problem by considering only a subset of the structures that G-rich DNA can adopt. Specifically, we calculated the number of ways to incorporate different sets of twelve G residues into a stable, three-tetrad, G4 core, while ignoring topology and non-canonical or higher-order interactions. We considered potential G4s with loop lengths of up to seven residues [44] and bulges with lengths of up to three [14]. For example, a hypothetical sequence G_3_TG_4_TG_3_TG_3_TG could adopt four different structures: GGGTGGGGTGGGTGGGTG, GGGTGGGGTGGGTGGGTG, GGGTGGGGTGGGTGGG(T)G, and GGGTGGGGTGGGTGGG(T)G, where G’s in the core are underlined and parentheses indicate bulged residues. Bulged G’s were not considered, since the non-bulged variant would always be more stable. We note that, whereas two-tetrad G4s exist, they are usually quite unstable in the absence of additional non-canonical interactions, which is beyond the scope of this analysis [28,49]. G4s containing more than three tetrads have been reported [29,50], although other studies suggest that strand-shifted conformations with 3-tetrad cores may be still be preferred, even when all G-tracts contain more than three G’s [51]. Furthermore, whereas we have restricted our analysis to a maximum loop length of seven, structures with larger loops are present in the human genome as well as other G4-forming sequences which do not fit our simplified model [3]. Thus in the following analysis, it should be remembered that we have considered an important but partial subset of the possible G4 structures formed by any given DNA sequence. The true number of accessible conformations is likely somewhat larger.

### 2.2. Predicting Stable G4 Structures

In order to realistically evaluate G4 polymorphism, it is important to account for the fact that some G4 structures are more likely to form than others. Previous studies have related experimental melting temperatures (T_m_) of G4s to the lengths of the loops (L_1–3_) [52] and the number (N_b_) and size (L_b_) of bulges [14]. Unsurprisingly, T_m_ decreases with increasing total loop length (L_1_ + L_2_ + L_3_) (Figure 1a). The dependence is reasonably well fit by a linear relationship; however, the prediction breaks down for the most stable G4s with the shortest loops. For example, the calculated melting temperature of the extremely stable (G_3_T)_3_G_3_ sequence (77.6 °C) indicated by the red circle in Figure 2a is actually lower by 6.4 °C than the measured melting temperatures of the longer-loop variants (84 °C and 81.8 °C, respectively), when the reverse should be true. Notably, when data for a single loop are examined in detail, T_m_ values show a curvilinear decrease with increasing L (Figure 2b), which is perhaps to be expected since theory suggests that the entropic cost of closing a loop during macromolecular folding varies as the logarithm of the length of the loop [53]. Plotting T_m_ as a function of log(L) produces far better linear relationships (Figure 2e). When data for all variants are plotted as a function of log(L_1_) + log(L_2_) + log(L_3_) (=log(L_1_L_2_L_3_)), a linear relationship is obtained (Figure 2d) with an improved R^2^ value compared to Figure 2a. The T_m_ predicted for (G_3_T)_3_G_3_ (at the y-intercept) is 6 °C higher than those determined experimentally for the longer-loop variants, as expected. The melting temperature can therefore be estimated for this dataset by the empirical equation
(1)$$T_{m}\left(L_{1},L_{2},L_{3}\right)=a-b\;\cdot \;log\left\{L_{1}L_{2}L_{3}\right\}$$
where *a* = 89.9 °C and *b* = 19.2 °C.

Stability data have also been reported for G4s containing different numbers of bulges of different lengths [14]. The introduction of a bulge leads to a roughly 20 °C reduction in T_m_, and each additional residue in the bulge reduces the T_m_ by about another 10 °C. We found that the data are well fit by the empirical relationship
(2)$$T_{m}\left(N_{b},L_{b}\right)=c-d\;\cdot \;N_{b}-f\;\cdot \;\left(L_{b}-N_{b}\right)$$
where N_b_ and L_b_ are the total number of bulges and the total number of bulged residues, respectively. For data obtained with 12 mM K^+^, *c* = 89.9 °C, *d* = 20.5 °C, and *f* = 8.5 °C (Figure 2c). For data obtained with 60 mM K^+^, *c* = 97.8 °C, *d* = 19.4 °C, and *f* = 8.5 °C (Figure 2f). Since the 12 mM K^+^ data and 60 mM K^+^ data produced similar values for *d* and *f,* we took the mean of these two parameters giving *d =* 20 °C and *f =* 8.5 °C. We combined Equations (1) and (2) to estimate the stabilities of all predicted G4s on the basis of their loop lengths, and the number and length of bulges according to:(3)$$T_{m}^{est}\left(L_{1},L_{2},L_{3},N_{b},L_{b}\right)=a-b\;\cdot \;log\left\{L_{1}L_{2}L_{3}\right\}-d\;\cdot \;N_{b}-f\;\cdot \;\left(L_{b}-N_{b}\right)$$

It should be kept in mind that the *T_m_^est^* parameter is not a precise predictor of the melting temperature, given the scatter evident in Figure 2a,d, and possible contributions of non-canonical and higher-order interactions in naturally occurring G4s [34]. Rather, a high *T_m_^est^* value indicates that a putative G4 structure has short loops and a small or no bulge whereas a low *T_m_^est^* value is indicative of long loops and/or bulges, with the values tied to physically reasonable melting temperatures.

### 2.3. Identifying G4 Regions (G4CRs) in DNA Sequences

To locate G4CRs within a given stretch of DNA, we identified all sets of 12 core G residues within the sequence that can theoretically form a G4 with T_m_^est^ ≥ 50 °C according to Equation (3), together with the accompanying loops (limited to ≤7 nt) and bulges (limited to ≤3 nt). Regions encompassing overlapping sets of Gs were defined as G4CRs. Figure 3 illustrates a hypothetical example in which the first G4CR (G4CR_1_) contains six G-tracts and can form G4s with eight distinct subsets of G residues. The second G4CR (G4CR_2_) is separate from the first because no stable G4 includes both the 3′ G-tract of G4CR_1_ and the 5′ G-tract of G4CR_2_. Note that for very G-rich sequences, the distinction between different G-tracts and between looped and bulged residues becomes somewhat blurred. For example, in G4CR_2_, the T is part of a loop in G4_1_, G4_2_, and G4_3_ and a bulge in the third G-tract in G4_4_ and G4_5_. The G immediately following the T is part of the fourth G-tract in G4_1_, part of a loop in G4_2_ and G4_3_, and part of the third G-tract in G4_4_ and G4_5_. Regardless, the locations and lengths of G4CRs can be assigned unambiguously.

Interestingly, the contributions of each G residue to the entire ensemble of G4s formed by the G4CR are quite different. For example, the Gs of the first tract in G4CR_1_ are only folded in two of the eight possible structures. In contrast, those of the third G-tract are folded in all eight of the possible structures. We refer to these relative contributions of each G as the folding multiplicity; values are listed for each G beneath the sequence in Figure 3. The values of multiplicity and the total number of G4s formed by a G4CR are related. Since each G4 is composed of exactly 12 G core residues in our simplified model, the total number of G4s is given by the sum of the multiplicities for all G residues in the G4CR, divided by 12.

### 2.4. Characteristics of G4CRs in Human Promoters

We analyzed portions of the human genome within −1999 and 2000 base-pairs of the transcription start sites (TSS) of 16,528 genes using the first promoter listed in the eukaryotic promoter database [54], considering both coding and non-coding strands of DNA. A cumulative plot of the incidence of G4CRs is shown in Appendix A, where the curves give the number of genes with fewer than a given number of G4CRs. As found previously for individual G4s, most (85%) genes contain at least one G4CR within the examined region [45]. The median number of G4CRs per gene (i.e., the value at the 50th percentile) is three, whereas about 1% of genes (154) have over 15 G4CRs. In some cases, these promoters contain G4CR-rich stretches that correspond to repetitive minisatellite DNA, as discussed below. We identified a total of 2847 G4CRs in this top 1% of genes. This is only about 5% of the total number of G4CRs we found in all genes (63,303), meaning that the statistics we report for G4CRs below are dominated by non-repetitive, non-satellite DNA. 

Figure 4a–d shows cumulative plots of the distributions of the lengths, G-content, the total number of G4 isomers (N_tot_), and total number of G4s that can form simultaneously in tandem (N_tand_). Data for G4CRs from the coding and non-coding strands were pooled since we did not observe any substantial differences between the two strands. The median length of a G4CR is about 25 nt, and 75% is shorter than about 32 nt (Figure 4a). However, a substantial number of G4CRs are much longer. Roughly 1% or about 600 G4CRs are longer than 71 nt, which is a considerable number when one recalls that this refers to a continuous stretch of nucleotides where each nucleotide has the potential to form part of a G4 structure. Figure 4e shows cumulative plots of the estimated melting temperatures of the G4s predicted to be formed within the G4CRs. The median T_m_^est^ is 60 °C, with about 10% predicted to melt above 70 °C. Identical T_m_^est^ curves were obtained for G4CRs of different lengths (Figure 4e). The guanine content of most G4CRs is between about 50% and 75%, although a few G4CRs were identified with 100% G-content, i.e., stretches of poly-guanosine (Figure 4b). Interestingly, different T_m_^est^ profiles were obtained for G4CRs with different G content, such that G4s located in G4CRs with higher G-content tended to have higher estimated melting points (Figure 4f). For example, the fraction of G4s with T_m_^est^ > 70 °C was about four-fold higher for G4s derived from G4CRs with >84% G than for those with ≤68% G. This implies that putative G4s from G4CRs with high G content have shorter loops and bulges compared to those from G4CRs with low G content. The median number of different G4 isomers (N_tot_) formed by a G4CR is four, but there are many with far greater degrees of polymorphism (Figure 4c). A total of 25% of G4CRs form greater than 14 different isomers, 1% form more than 179 G4 isomers, whereas 20 G4CRs can potentially form over 1000 distinct G4 structures each. Similarly to what was seen with G-content, T_m_^est^ values are higher for G4s derived from G4CRs with larger N_tot_, compared to those with lower N_tot_ (Figure 4g). In other words, G4s from G4CRs with greater degrees of polymorphism tend to have shorter loops and bulges. The overwhelming majority (>95%) of G4CRs can only form one G4 structure at a time (N_tand_ = 1), as shown in Figure 4d. About 1% can form three or more tandem G4s and 0.1% (or about 100 G4CRs) can form five or more at one time. We did not observe any difference in the predicted stabilities of G4s derived from G4CRs with different N_tand_ values (Figure 4h).

We next explored the correlation between the G4CR characteristics by constructing scatter plots for every pair of the four parameters discussed above (Figure 5). Surprisingly, apart from the strong and expected increase in the number of tandem G4s with increasing G4CR length (Figure 5b), correlations among the other parameters were weak at best. For example, G4CRs with between about 25 and 60 nt in length can have anywhere from 2 to over 500 isomers (Figure 5a). Thus, the number of G4 isomers formed by a G4CR does not depend strongly on the length of the region. The very highest G contents were observed for the shortest G4RCs, implying that stretches of nearly 100% guanosine are limited to about 15–25 nt in length (Figure 5c). The lowest contents of G (≈40%) were observed for G4CRs about 40 nt in length. Conversely, the very longest G4CRs of several hundred bases (Figure 5c) and those with the largest number of tandem G4s (more than 10, Figure 5f) had intermediate G contents of 60–80%. These intermediate G content G4CRs formed anywhere between 1 and >1000 G4 isomers (Figure 5e). Low G content (<50%) led to small N_tot_ values, likely due to a lack of G residues to form alternative structures. Very high G content (>90%) was associated with moderate N_tot_ values (<100), likely because these G4CRs tended to be short. The fact that the key characteristics of G4CRs are largely uncorrelated implies that G4CRs can be classified into multiple categories: short vs. long, low vs. high G content, and small vs. large numbers of isomers (N_tot_) (relative to the median values). Note that longer G4CRs are strongly associated with a larger number of tandem G4s, so N_tand_ does not represent a separate parameter for the purposes of categorization. This rough division gives eight different classes of G4CR. It is an interesting question to which extent these different classes have divergent biophysical properties, functional relevance, or biological roles.

### 2.5. Distribution of G4CRs in Gene Promoters

It is well known that the distribution of putative G4 forming sequences within the human genome is distinctly non-random. G4 motifs are enriched adjacent to transcription start sites (TSSs) in gene promoters from humans [44,45,46] and a wide range of other eukaryotes [55], with asymmetric densities on coding vs. non-coding strands, pointing to a general regulatory role for G4s [56]. Plots of G4 propensity calculated using the quadparser, G4Hunter, and QPARSE algorithms all produce characteristic sharp peaks about 100 nt immediately upstream from the TSS, equally sharp dips in the 100 nt immediately following the TSS, and broader peaks over the next roughly 500 nt [44,45,46]. The first peak and dip are evident on both coding and non-coding (template) strands, whereas the last peak is far more prevalent on the coding strand compared to the template strand. We set out to evaluate the extent to which the distribution of G4CRs matches these previous results and whether the distributions are correlated with the degree of polymorphism or other G4CR characteristics. We first calculated the fraction of genes in which a given position relative to the TSS lies within a G4CR bearing certain characteristics of length, G content, etc. for a region extending from −1999 (upstream) to +2000 (downstream) of the TSS of all 29,598 promoters listed in the eukaryotic promoter database [54]. We then normalized the distributions to facilitate comparisons between G4CRs with differing characteristics. This was completed by setting the sum of probabilities over all 4000 positions in the calculation window equal to 1, which accounts for the fact that the likelihoods of lying within longer and/or more common types of G4CRs (i.e., with characteristics close to the mode) are larger overall. Figure 6 shows the normalized probability distributions for G4CRs for coding (Figure 6a–d) and non-coding (Figure 6e–h) strands. All of the plots closely mirror what was reported using the quadparser, G4Hunter, and QPARSE algorithms with sharp peaks upstream on both coding and non-coding strands and broad peaks downstream of the TSS, particularly on the coding strand. Interestingly, the sharpness of the distributions varies depending on the characteristics of the G4CR. For example, Figure 6a,e shows the normalized probabilities of lying within G4CRs of different lengths for the coding (Figure 6a) and non-coding (Figure 6e) strands. The pre-TSS (left) peaks are substantially sharper for longer G4CRs than for shorter ones, with the highest peak for G4CRs with length > 69 nt followed by those 31 < length ≤ 69 nt. As a result, the enrichment in G4CRs in the −200 to 0 relative to −1500 to −1300 regions (relative to TSS) is 11-fold for G4CRs longer than 69 nt compared with only five-fold for G4CRs less than 25 nt. In other words, longer G4CRs are more tightly clustered immediately upstream from the TSS, compared to shorter ones. Figure 6b,f shows similar data for G content. In this case, G4CRs with intermediate %G values (69–73% and 74–84%) exhibit the highest pre-TSS peaks in probability. In terms of polymorphism, G4CRs with the highest total number of G4 isomers (>168 and 14–168) show the sharpest clustering prior to the TSS, compared to G4CRs with fewer than 13 isomers, whereas G4CRs with two and three tandem G4CRS show a higher peak than those with more than three or just a single isomer. The greater clustering of long G4CRs with high numbers of G4 isomers that form two or more contiguous G4s points to a relationship between these characteristics and G4 biological function.

We then extended this analysis by considering promoter regions from a variety of organisms including the vertebrates *M. mulatta* (monkey), *M. musculus* (mouse), *R. norvegicus* (rat), *C. familiaris* (dog), *G. gallus* (chicken), and *D. rerio* (fish), the invertebrates *D. melanogaster* (fly), *A. mellifera* (bee), and *C. elegans* (worm), the plants *A. thaliana* (thale cress) and *Z. mays* (corn), the yeasts *S. cerevisiae* and *S. pombe*, and the parasitic protozoan *P. falciparum*. Many of these organisms have been previously analyzed using the G4Hunter algorithm and classified as containing a high density (human, monkey, mouse, rat, dog, and chicken), an intermediate density (fly, bee, and fish), a low density (yeasts, worm, and thale cress), or very low density (protozoan) of G4-forming sequences. We obtained similar results, with an average of 3.6 G4CRs identified per promoter in the high G4 density group, 0.19 G4CRs per promoter in the intermediate G4 density group, 0.08 G4CRs per promoter in the low G4 density group, and 0.005 G4CRs per promoter in the protozoan (Appendix A). We found that corn (which was not examined by G4Hunter) has a relatively high abundance of G4CRs (0.96 per promoter). This is intriguingly similar to rice, which was included in the high G4 density group along with mammals and chicken, in the G4Hunter study. We next examined distributions of G4CRs relative to the TSS for the species listed above. For all of the high G4-density vertebrate species, we observed clustering of G4CRs similar to that in humans (Appendix A). On both coding and non-coding strands, distributions show a large peak just upstream of the TSS, followed by a sharp dip at the TSS, and a second peak just downstream. In monkeys, the longest and most polymorphic G4CRs clustered more tightly near the TSS than shorter and less polymorphic ones, as we saw for humans. For other high G4-density organisms (mouse, rat, dog, and chicken) this trend was still present, albeit to a slightly lesser extent. For these species, the shortest and least polymorphic 50% of G4CRs were less clustered near TSS than longer and more polymorphic ones. However, unlike humans and monkeys, the most tightly clustered G4CRs were not necessarily the top 1% of G4CRs in terms of length or polymorphism. For species with fewer G4-rich genomes, essentially no clustering of G4CRs was observed near the TSS at all, similar to what we observed with randomly shuffled human promoter sequences (Appendix A). Thus, in animal genomes with a high propensity to form G4s, G4CRs are highly enriched near the TSS, and longer and more polymorphic G4CRs are more enriched than shorter and less polymorphic ones. Interestingly, in corn, the distribution of G4CRs was very different than that seen in animals. Clustering was much more pronounced on the non-coding strand compared to the coding one, and the distribution had a single peak, with no sharp dip at the TSS (Appendix A).

### 2.6. Analysis of G4CRs with Validated Biological Activity

There are several oncogene promoters where ample experimental evidence exists to show that G4 formation is correlated with gene expression. It is therefore of interest to examine in some detail the G4CRs from these genes (Appendix A) and compare their characteristics to those of G4CRs, in general. A useful tool in this regard is the calculation of multiplicities, i.e., the number of distinct G4 structures that include a particular G residue in the core, as these provide a measure of polymorphism at the single nucleotide level. As shown in Appendix A, about 20% of Gs in G4CRs have a multiplicity of one, which matches our observation that 18% of G4CRs form only a single G4 structure. The median multiplicity of G residues in G4CRs is four, whereas about 1% of G’s have multiplicities of over 100.

We first examined the promoter region of the *MYC* proto-oncogene, which is overexpressed in more than 50% of cancers [57]. The region −142 to −115 upstream of the TSS on the non-coding strand of DNA has been shown to fold into a parallel G4 structure [13,20,39]. Disruption of the G4 by mutation was shown to increase the expression of a reporter gene under the control of the *MYC* promoter by about three-fold. Conversely, the addition of a G4-stabilizing ligand decreases *MYC* gene expression, only when the promoter contains the G4 element [39]. Our analysis identified six G4CRs in the *MYC* promoter, two on the coding strand, and four on the non-coding strand. The previously studied G4 is contained in G4CR n2, which is the longest of the six (at 59 nt), contains the largest number of G4 isomers (at 85), and the highest multiplicity values (at 70), and has the G4s with the highest values of *T_m_^est^* (at 84 °C). The G4 isomer whose structure was solved by NMR spectroscopy and is commonly referred to as the “biologically relevant” conformation is predicted to have the highest melting temperature of the 85 isomers (tied with three others). Interestingly, the G4CR encompasses a longer region than is typically studied and allows a maximum of two G4s to form simultaneously. A recent report investigated this longer region experimentally, concluding that two G4s can, in fact, fold in tandem [37].

We next examined *VEGFA*, which is overexpressed and promotes tumor survival, growth, and metastasis in a range of human cancers [58,59]. A region 50–85 nt upstream of the TSS forms a parallel G4 [60,61]. It is essential to *VEGF* expression [62], recruiting the transcription factor Sp1, which binds tightly to both duplex and G4 conformations [63]. Conversely, G4-binding ligands suppress *VEGF* expression [64]. We identified 12 G4CRs in the *VEGFA* promoter. The one corresponding to the functional region (c3) is the second longest (30, as opposed to 31 nt for n1), has the second most G4 isomers (19 as opposed to 20 for c1), and has the third-highest maximum *T_m_^est^* (78, as opposed to 81 for c1 and n6). Notably, these other regions (c1, n1, and n6) are all more than 1 kb distant from the TSS. Of the 19 isomers we identified for the G4CR c3, the one we predicted to be the most stable is the one observed experimentally.

For *BCL2*, whose overexpression is linked to a large variety of cancers [65], deletion of a G4-forming region immediately before the TSS increases promoter activity [66], and when this region is placed upstream from a reported gene, G4-disruptive mutations increase expression while G4-binding ligands reduce it [31]. We found six G4CRs in the *BCL2* promoter region. G4CR 3n, which corresponds to the previously identified region, is by far the longest (at 80 nt), has the by far largest number of G4 isomers (at 40), and has the G4 with the highest *T_m_^est^*. However, the dominant conformations determined in the solution have long (>10 nt) hairpin loops, which are not captured by our algorithm [31,65,67]. The proto-oncogene *KIT*, which is associated with a large number of human cancers [68], has previously been found to contain three adjacent G4s, containing three, two, and three G-tetrads, respectively [21,69]. Reporter gene assays have shown that disruptive mutations of the first G4 elevate gene expression, whereas in the second two, expression is suppressed, likely due to reduced recruitment of the transcription factor Sp1 [21]. Furthermore, G4-stabilizing ligands reduce *KIT* expression in carcinoma cell lines [70]. We identified only 3 G4CRs in the *KIT* promoter region. G4CR c1 encompasses the first G4 and part of the second, which with only 2-tetrads is not selected by our algorithm. This G4CR has a length (33), N_tot_ (14), and maximum *T_m_^est^* (76 °C) on par with the other functionally validated G4CRs examined here. The experimentally determined structure of the first G4 matches the most stable isomer identified for G4CR c1 [71]. The G4CR c2 corresponds exactly to the third, previously studied G4, and its G4 isomer predicted to be the most stable by our algorithm matches the structure determined experimentally [69]. The three G4s of the *KIT* promoter are believed to be stabilized by higher-order stacking interactions that are not captured by our algorithm [21,35].

The promoter of *KRAS* contains two G4-forming regions, termed the near-G4 and mid-G4 [72,73] (A far-G4 region exists, but does not, in fact, form a stable G4 structure). The near-G4 region has been shown to recruit nuclear factors affecting gene expression [74]. In addition, the binding of G4-ligands to the *KRAS* promoter decreases the expression of a reporter gene, an effect that is primarily mediated by the mid-G4 region, rather than the near-G4 [75]. Interestingly, the G4CR n1, which corresponds to the mid-G4, is longer (52 vs. 26 nt), more polymorphic (25 vs. 2 isomers), and has a higher maximum *T_m_^est^* (65 vs. 58 °C) compared to G4CR n2, which corresponds to the near-G4. Considering all 62,791 G4CRs we identified in all gene promoter regions, the median length of a G4CR is 25, and median N_tot_ value is four. Thus, among these biologically validated promoter G4s, there appears to be an over-representation of longer and more polymorphic G4CRs that contain G4s with shorter loops and fewer bulges. Applying this lesson to *KRAS*, the longest (62 nt), most polymorphic (103 isomers), and most stable (max *T_m_^est^* 75 °C) G4CR we identified is located about 300 nt downstream from the TSS on the coding strand (G4CR c1), and would therefore be transcribed into mRNA. There is already some evidence for mRNA G4 regulation of *KRAS* translation, involving several two-tetrad structures [76], but to our knowledge, the G4CR c1 has not yet been investigated. We would conclude that the characteristics of this G4CR would identify it as being a particularly interesting candidate for future study.

### 2.7. The GReg Webserver

We have made our **G**4-Containing **Reg**ion (GReg) algorithm available as a webserver on our lab’s webpage (https://www.mcgill.ca/mittermaierlab/greg-webserver). This website allows users to enter an unlimited number of DNA sequences up to 32 k nt in length. It provides both graphical and text output describing multiplicities (similar to Figure 7), lengths, positions, G-content, N_tot_, and N_tand_ of all G4CRs present in the inputted sequences, as well as detailed listings of every putative G4 formed in each G4CR and its T_m_^est^. Instructions for using the webserver are included in the Appendix A.

## 3. Discussion

Polymorphism has long been recognized as an intrinsic property of G4s. In fact, some of the earliest algorithms (as well as later ones) used to enumerate G4s in genomic data explicitly recognized and eliminated multiple structures involving the same region of Gs to avoid overcounting [44,45,46]. Similarly, mutations are routinely used to trap single conformations and eliminate unwanted dynamics in structural studies [47]. However, these approaches discard a great deal of information on the nature, prevalence, and potential roles of G4 polymorphism. To some extent, the discussion thus far has been guided by our tendency to refer to the G4 structure as the biologically active unit. In cases with low polymorphism, it makes intuitive sense to assign a single conformation as the “biologically relevant” one and account for simple dynamics in terms of “G-register exchange” [17] or “spare tire” motions [22]. However, nature provides many examples that do not fit neatly into this kind of single-structure, single-function paradigm. An extreme example of this is minisatellite DNA, also known as variable number tandem repeats (VNTRs). These are repeating tandem units of more than two nucleotides that can be repeated several hundred times [77,78]. The number of repeats differs from individual to individual and VNTRs are useful as genetic markers [77,79]. When the repeating units are G-rich, VNTRs can form G4s. In fact, several characterized G4 structures are derived from short regions of VNTR DNA [23,24]. In their entirety, VNTRs can produce a rich diversity of structures, in part because G-tracts repeated more than about 20 times in tandem can lead to a frustrated energy landscape with an enormous number of different folded G4 forms [25]. Furthermore, G4 folding has been linked to the genetic instability of VNTRs [80,81,82] whereas the number of repeats in VNTRs located in gene promoter regions is related to the expression levels of the corresponding proteins [83], echoing the influence of G4 folding on gene expression. In examples such as these, it makes little sense to ascribe the biological activity to a single-folded G4. Instead, the concept of the G4CR, as defined in this study, provides an alternative definition of the biologically relevant unit that can be rigorously defined even when the structural folding landscape becomes exceedingly complex. In fact, our algorithm picked out several G4CRs that have previously been identified as VNTRs [84,85,86]. Notably, there is no clear division between short G4CRs that form a unique G4 and those that form thousands. The G4CRs we identified present continuums of length and polymorphism spanning several orders of magnitude. The concept of the G4CR provides a common framework for understanding the role of genomic G4s in all the different contexts in which they appear.

Our systematic survey quantitatively confirms the picture that has been emerging from studies of dozens of G4s with multiple isomers; polymorphism is a ubiquitous feature of G4 folding. In fact, only a minority of G4CRs (≈20%) contain a solitary G4 structure (Figure 4c). Even relatively short G4CRs that form only one G4 structure at a time can adopt up to hundreds of different folds sequestering different subsets of G residues in the core (Figure 5d). Furthermore, there are hints that higher degrees of polymorphism are related to biological function. We found that enrichment immediately upstream of the TSS is greatest for G4CRs that are longer and contain greater numbers of G4 isomers (Figure 6a,c). As well, some of the best-studied G4CRs with validated activities in controlling gene expression are longer and more polymorphic than the median, as discussed above. This result opens up new possibilities for uncovering biologically relevant sequences. More research is needed to clarify the relationships between G4CR characteristics and activity, however, it already seems that longer, more polymorphic G4CRs may be good candidates for deeper study. As well, our bioinformatic search has uncovered a sizeable number of interesting regions that defy the paradigm of a single biologically relevant isomer. These have hundreds to thousands of different isomers, leading to a situation where polymorphism itself may play more of a role in governing the biophysics and activity of these DNA regions than the three-dimensional structure of any single isomer. Such highly polymorphic sequences are found in promoters of genes implicated in different forms of cancer and are listed in Table 1. Furthermore, some of the sequences we identify are unusually G rich (up to 83%) and unusually long (up to 369 nt). The complementary strands, therefore, contain regions that are equally C-rich and equally long. Previous work has shown that i-motifs, four-stranded structures formed by C-rich DNA, are generally unstable at physiological pH [87], but that longer and more C-rich sequences fold more readily [88]. I-motifs have putative roles in controlling gene expression, similarly to G4s [87]. Thus, these exceptionally C-rich regions in gene promoters are highly interesting in their own right.

It is useful to consider exactly how polymorphism may impact G4 function. We note that the isomers predicted by our algorithm are likely not equally populated. In most cases, the ensemble of structures will be dominated by the one or several most stable isomers, whereas many of the less stable ones may only be present at levels of a fraction of a percent at equilibrium. Nevertheless, the existence of many less-stable isomers can still be functionally relevant. For example, we previously characterized G-register exchange among four structural isomers in a portion of the main G4CR from the *MYC* promoter. We found that even though a single isomer accounted for as much as 80% of the folded ensemble, the presence of three additional weakly populated isomers doubled the apparent folding equilibrium constant and increased the effective melting temperature by 3.4 °C due to entropic stabilization of the folded state [17]. Furthermore, the existence of the minor states increases the apparent folding rate by a factor of 2.5 [19], since the folding of each of the four isomers represents a separate and parallel pathway to the folded state. This is particularly relevant to situations where G4 function relies more on kinetics (rapid folding) than on thermodynamics (high stability) [35]. Although these effects are modest in the case of *MYC*, there are many G4CRs with orders of magnitude more isomers. Determining how the effects scale with the number of isomers is an interesting avenue for future research, as the impact on highly polymorphic G4CRs could be substantial. The existence of multiple weakly populated isomers is also relevant to the resilience of DNA to oxidative damage. Oxo-guanine residues located in a G4 core are not always accessible to base excision repair enzymes [96]. However, it has been shown that a fifth G-tract, if present, can replace the damaged G-tract, hence the “spare tire” terminology. This extrudes the oxo-guanine into a long loop where it becomes a substrate of the repair machinery [97]. Some of the polymorphisms we have calculated using our algorithm fall explicitly into the category of spare tire dynamics with one complete G-tract replacing another. Other types of isomerization events might represent cryptic spare tire motions as well. In fact, the transition between any two isomers identified by our algorithm extrudes at least one G residue into a loop or flanking region, by definition. For example, the hypothetical oxidized G4 GGGG**oG**TGGGTGGGTGGG could isomerize to place the damaged G residue in a loop (GGGG**oG**TGGGTGGGTGGG), without directly replacing one G-tract with another. Whereas the relationship between G4 structure and accessibility to DNA repair enzymes is not yet fully understood [98], access to larger numbers of alternative structural isomers can be considered potentially protective against oxidative damage. As well, our analysis has shown that both higher G content and larger numbers of isomers are statistically correlated with the presence of more stable G4s (i.e., those with shorter loops and bulges, Figure 4b,c). We have examined this relationship in slightly more detail, comparing the distribution of T_m_^est^ values as simultaneous functions of both %G and N_tot_. We find that G4CRs with lower G content and fewer isomers than the medians (blue in Figure 8a,b) have lower T_m_^est^ values than those with lower G content and more isomers (yellow). Similarly, G4CRs with higher G content and fewer isomers (orange) have lower T_m_^est^ values than those with higher G content and more isomers (purple). Thus higher levels of polymorphism are directly correlated with G4s having shorter loops and bulges. We speculate that G-rich, highly polymorphic sequences are more evolutionarily accessible than unique, highly stable sequences such as (G_3_T)_3_G_3_, implying that the evolutionary path towards stable G4s creates a large number of additional G4 isomers. Overall, both the functional (entropic stabilization, parallel folding pathways) and collateral (more stable individual G4 isomers) explanations are mutually consistent and both may be at play in explaining the ubiquity of G4 polymorphism.

It must be noted that our algorithm is quite conservative in the identification of G4 isomers. For example, it does not consider stabilizing interactions outside the G4 core, such as the stacking of bases [32], the hairpin formation in loops and bulges [30,31], or higher-order G4/G4 interactions [33]. This inevitably led us to discard G4 structures as unstable that might form readily in reality. We ignored G4s with only two tetrads [28], although these can be stabilized considerably by non-canonical interactions, and we discarded structures with loops longer than 7 nt, although much longer loops are sometimes observed in stable G4s [3,52]. We also disallowed bulges containing G residues, since the isomer with the G in the core would be expected to be far more stable, even though bulged Gs are possible in principle and could be relevant in cryptic spare tire dynamics, as discussed above. The flip side is that all the isomers we predict are likely to be stable. Their presence may be obscured by more energetically favorable isomers that make up the bulk of the ensemble. However if all non-core G residues were replaced with mimics such as inosine [17,19], it is highly probable that the 12 G residues predicted to be in the core would fold into a G4 with three tetrads. Thus, the very large numbers of isomers we calculate for many of the G4CRs likely underestimate rather than exaggerate the true number.

Additionally, given the prevalence of G4s in telomeres [99] and the promoter regions of oncogenes, and their ability to affect expression levels, targeting specific G4s with small molecule drugs has been an attractive new avenue towards developing cancer therapeutics [56]. However, the fact that most G4CRs can adopt multiple structures raises some fundamental questions about how this is best achieved. Selectivity is already an issue in targeting particular G4s since, unlike proteins, G4s possess similar cores and differ primarily in the identity of the loop residues. Even if a drug achieves specificity for a particular G4 with a unique three-dimensional structure, what is the likelihood that the target adopts alternative isomers and escapes binding? Conversely, to what extent might one of the many isomers of an off-target G4 bind the drug? Interfaces between tandem G4s have been proposed as more specific drug targets [36,37]. However, even here, polymorphism introduces complications. There are far fewer ways to fold a G4CR with the maximum number of tandem G4s than there are with a smaller number of G4s [25]. Therefore, most of the possible isomers formed by a G4CR lack the targeted interface, although this is potentially compensated by stabilizing higher-order interactions which would increase the relative populations of the tandem structures. Thus, analyzing drug targets in terms of polymorphic G4CRs rather than as unique G4 structures seems a surer way to account for the complexity of this challenge.

Ultimately, the bioinformatic tools we have developed here and made publicly available on the GReg webserver will facilitate a better understanding of how G4 polymorphism intersects with biological function and evolution. They simplify the identification of longer and more polymorphic G4CRs, which we have found are positively associated with biological activity. They make it easy to identify the boundaries of G4-containing regions, thereby ensuring the full sequence can be analyzed. For example, our algorithm identified a longer region in the *MYC* promoter than had been typically studied; this longer region was just recently found to have structural relevance [37]. By mapping out the polymorphic landscape of G4CRs, our tools will make it easier to predict and interpret the effects of genetic variation on G4 formation, both between species and between individuals. This same information may also help us to better direct G4 ligands specifically to one stretch of G4-containing DNA over another. Finally, this bioinformatic analysis has underlined some fundamental unanswered questions regarding G4-containing regions. Why do some putative G4s occur in isolation and form a single unique structure, while others occur in the context of contiguous G4-forming regions hundreds of nucleotides long with potentially many thousands of alternate folded states? How do the biophysical properties of these various types of G4-containing regions differ, and how does this impact biological function? We hope that the bioinformatic tools reported here will serve as a stepping stone toward answering these questions and developing a deeper biophysical understanding of G4s and their activity.

## 4. Materials and Methods

### 4.1. The GReg Algorithm

#### 4.1.1. Generating Possible G4 Sequences (get_GQs)

Initially, a series of matrices (G4_matrix_) were generated that encoded, in their rows, every possible G4 configuration, where core G positions were indicated by values of 1, loop positions were assigned 0, and bulged positions were assigned a value greater than 12 (13 was used here). The number of columns of a particular G matrix was given by the total number of loop and bulged residues plus 12 (core positions). We used a maximum loop length of 7 and bulge length of 3. Only one bulge was considered since two or more bulges reduced the T_m_^est^ below the selected threshold of 50 °C. Thus 21 different G matrices were generated, containing between 15 columns (for three loops of 1 and no bulge) to 36 columns (for three loops of 7 and a bulge of 3). The rows were generated combinatorically, combining every possible length of the three loops with every possible bulge position and length. This gave a total of 8575 different sequences: 7^3^ × 8 × 3 + 7^3^ for (three loops of 1–7 residues) × (two possible bulge locations per G-tract) × (bulge lengths of one to three) + (the possible G4 sequences with no bulge). Next, the T_m_^est^ corresponding to each row of each G4_matrix_ was calculated using Equation (3), and only rows with T_m_^est^ ≥ 50 °C were kept for further analysis resulting in 699 different G4 motifs. Thus, each row of each G4_matrix_ corresponded to a pattern of core, loop, and bulged residues that would produce a G4 structure with an estimated melting temperature greater than 50 °C. For example, a hypothetical G4_matrix_ with 18 columns is shown below. The first row corresponds to a G4 with no bulges and loop lengths of (1,3,2). The second row corresponds to a G4 with loop lengths of (1,1,2) and a bulge of two residues between the second and third G in the second G-tract. The third row corresponds to a G4 with no bulges and loop lengths of (1,4,1).
$$G4_{matrix}=\left(\begin{array}{c} 1\;1\;1\;0\;1\;1\;1\;0\;0\;0\;1\;1\;1\;0\;0\;1\;1\;1\\ 1\;1\;1\;0\;1\;1\;\left(13\right)\;\left(13\right)\;1\;0\;1\;1\;1\;0\;0\;1\;1\;1\\ 1\;1\;1\;0\;1\;1\;1\;0\;0\;0\;0\;1\;1\;1\;0\;1\;1\;1\\ \vdots \end{array}\right)$$

#### 4.1.2. Converting Original DNA Sequence (convert_Seq)

Next, the DNA sequence of interest was converted into a column vector wherein all G residues were indicated by 1s and all other residues were indicated by 0s. Note that this encoding is different from the one used in the G4_matrix_ above. The sequence was then split into possible G4CRs (pG4CRs), by removing all stretches of 0s longer than the maximum loop length (7), and retaining the intervening regions. Any pG4CRs with a sum of less than 12 were discarded, as these contain fewer than 12 G residues. For example, the following DNA sequence:T C T A C A A A G G G T G G G A G T G G G G T G G G T A T C T C A T G G A G C T C T T A C A
would be split into 2 pG4CRs:$$\begin{array}{c} pG4CR_{1}={\left[1\;1\;1\;0\;1\;1\;1\;0\;1\;0\;1\;1\;1\;1\;0\;1\;1\;1\right]}^{T}\\ pG4CR_{2}={\left[1\;1\;0\;1\right]}^{T} \end{array}$$
where the second pG4CR would be immediately discarded.

#### 4.1.3. Analyzing Possible G4CRs (analyze_Seq)

The number of possible G4 structures in each pG4CR was evaluated by matrix multiplication with each G4_matrix_ in turn. A sliding window with the same number of elements as the number of columns in the G4_matrix_ was extracted from the pG4CR and multiplied by the G4_matrix_, producing a column vector (GReg_vector_) with the same number of rows as the G4_matrix_. The value of each element of the GReg_vector_ indicates whether or not the window contains G residues at all core positions indicated by the corresponding row in the G4_matrix_. A value of 12 indicates that Gs are present at all core positions. A value less than 12 indicates that some core positions do not contain G residues in the window. A value greater than 12 indicates that a G is present in a bulged position, as illustrated below.
$$G4_{matrix}\cdot pG4CR=GReg_{vector}$$
$$\left(\begin{array}{c} 1\;1\;1\;0\;1\;1\;1\;0\;0\;0\;1\;1\;1\;0\;0\;1\;1\;1\\ 1\;1\;1\;0\;1\;1\;\left(13\right)\;\left(13\right)\;1\;0\;1\;1\;1\;0\;0\;1\;1\;1\\ 1\;1\;1\;0\;0\;0\;1\;1\;1\;0\;0\;1\;1\;1\;0\;1\;1\;1\\ \vdots \end{array}\right)\cdot {\left[1\;1\;1\;0\;1\;1\;1\;0\;1\;0\;1\;1\;1\;1\;0\;1\;1\;1\right]}^{T}=\left(\begin{array}{c} 12\\ 25\\ 11\\ \vdots \end{array}\right)$$

In the hypothetical example above (in which the length of the pG4CR happens to be equal to the width of the G4_matrix_), only the (1,3,2) loop isomer (top row) is counted as a G4 structure in the pG4CR. The pG4CR contains G residues at bulge positions for the (1,1,2) loop isomer (second row) resulting in a value of 25. One core position in the (3,2,1) loop variant (third row) did not correspond to a G in the pG4CR, leading to a GReg_vector_ element of 11. The analysis was repeated with the sliding window incremented across the entire pG4CR and for each G4_matrix_. The total number of G4 isomers formed by a pG4CR was calculated as the total number of GReg_vector_ elements that are equal to 12, summing over all positions of the sliding window and all G4_matrix_s. pG4CRs that did not produce a single GReg_vector_ element equal to 12 were discarded. pG4CRs which produce multiple GReg_vector_ elements equal to 12, but which had sections of non-overlapping G4 motifs were split into separate G4CRs. The multiplicity of each guanine in the pG4CR was calculated by aligning each selected row of each G4_matrix_ with the original sequence and summing over the columns.

### 4.2. Analysis of Human Promoters

The locations of all human promoters were downloaded from the Eukaryotic Promoter Database [54]. The promoter sequences were extracted from the GRCh38 build of the human genome with a window of −1999 to 2000 bp surrounding the transcription start site. For the analysis of G4CRs in the human genome, 16,528 unique promoters were analyzed using the first promoter labeled on the eukaryotic promoter database. Both coding and non-coding strands were analyzed together. Redundant promoters were not analyzed to avoid overcounting G4CRs which appeared multiple times. When analyzing the positional dependence of G4CRs all 29,598 promoters were analyzed, and coding and non-coding strands were analyzed separately.

### 4.3. Scripting

The GReg algorithm and genome-wide searches were performed using in-house MATLAB scripts, using MATLAB 2021a. Commented examples of each script for the GReg algorithm, an intuitive GUI for the GReg algorithm, and the python code used on the GReg webserver can be found at https://github.com/Christopher-Hennecker/GReg.

## Figures and Tables

**Figure 1 ijms-23-16020-f001:**
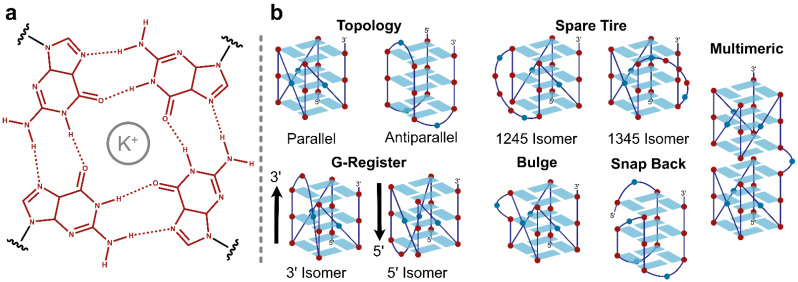
(**a**) Hydrogen bonding pattern of a G-tetrad in the G4 core. (**b**) Representative structural features of G4s: topological isomers, spare tire isomers, G-register isomers, bulges, snap back structures, and multimeric G4s.

**Figure 2 ijms-23-16020-f002:**
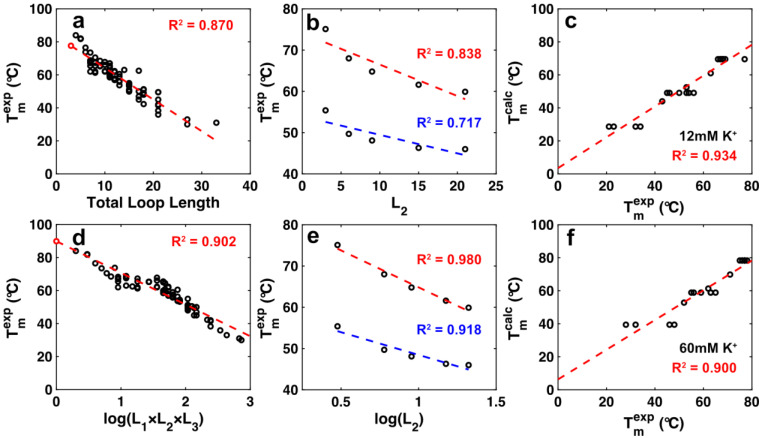
(**a**) Dependence of experimental T_m_ on total loop length. (**b**) Dependence of experimental T_m_ on the length of L_2_ for L_1_,L_3_ = TTT (blue) and L_1_,L_3_ = T (red). (**c**) Correlation between experimental and predicted T_m_ values for bulges of different lengths in 12 mM potassium according to Equation (3). (**d**) Dependence of experimental T_m_ on the logarithm of the product of the loop lengths. (**e**) Dependence of experimental T_m_ on the logarithm of the product of the loop lengths for L_1_,L_3_ = TTT (blue) and L_1_,L_3_ = T (red). (**f**) Correlation between experimental and predicted T_m_ values for bulges of different lengths in 60 mM potassium according to Equation (3). Data in (**a**,**b**,**d**,**e**) taken from [52]. Data in (**c**,**f**) taken from [14].

**Figure 3 ijms-23-16020-f003:**
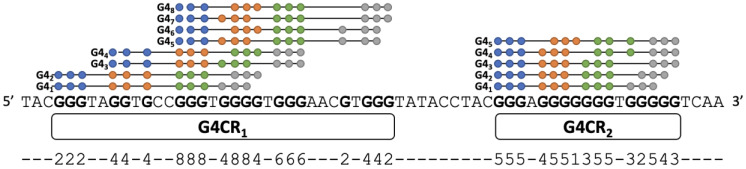
Hypothetical stretch of DNA containing two G4CRs. Filled circles above the sequence indicate the core G residues in each possible isomer. The multiplicity of each G residue is indicated at the bottom of the figure.

**Figure 4 ijms-23-16020-f004:**
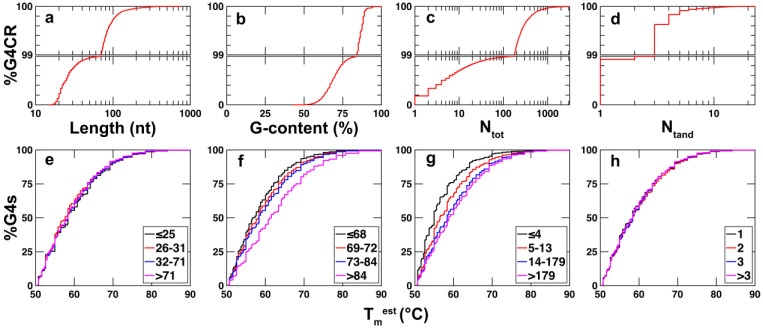
Cumulative plots of the (**a**) length, (**b**) G-content, (**c**) N_tot_, (**d**) N_tand_ of G4CRs found in the human promoters. Cumulative plots of the estimated melting temperatures for G4s derived from G4CRs binned by (**e**) length, (**f**) percentage of residues that are G (G-content), (**g**) total number of G4 isomers (N_tot_), (**h**) maximum number of simultaneous tandem G4s (N_tand_).

**Figure 5 ijms-23-16020-f005:**
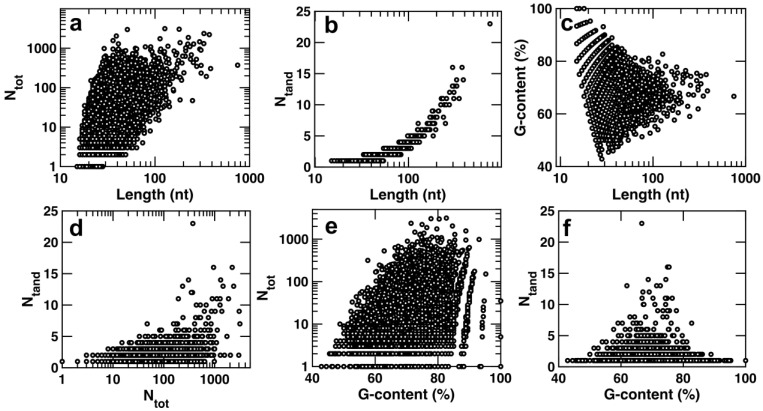
Scatter plots of each of the different features of G4CRs found in this study: length, percentage of residues that are G (G-content), total number of G4 isomers (N_tot_), maximum number of simultaneous tandem G4s (N_tand_). (**a**) N_tot_ vs. Length. (**b**) N_tand_ vs. Length. (**c**) G-content vs. Length. (**d**) N_tand_ vs. N_tot_. (**e**) N_tot_ vs. G-content. (**f**) N_tand_ vs. G-content.

**Figure 6 ijms-23-16020-f006:**
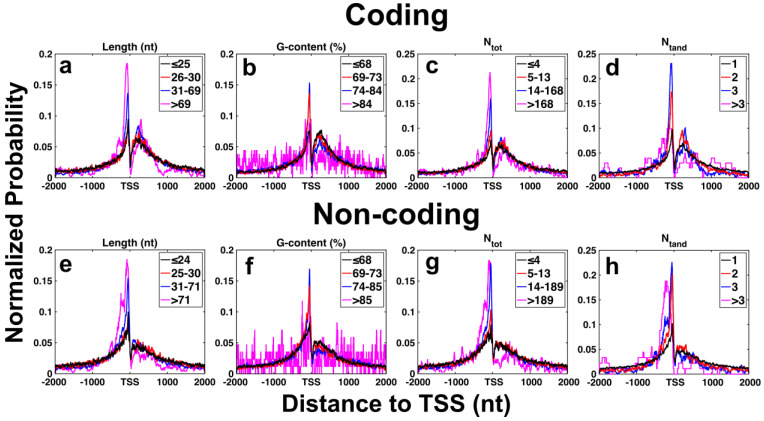
Likelihood that a residue lies within a G4CR possessing certain characteristics, plotted as a function of the residue’s position relative to the transcription start site (TSS), for human gene promoters. Colors represent the bottom 50% of G4CRs (black), 50–75% of G4CR, 76–99%, and top 1% of G4CRs. Specific values are given in the legend of each panel. Panels (**a**–**d**) are for the coding strand (length, G-content, N_tot_, and N_tand,_ respectively). Panels (**e**–**h**) are for the non-coding strand (length, G-content, N_tot_, and N_tand,_ respectively).

**Figure 7 ijms-23-16020-f007:**
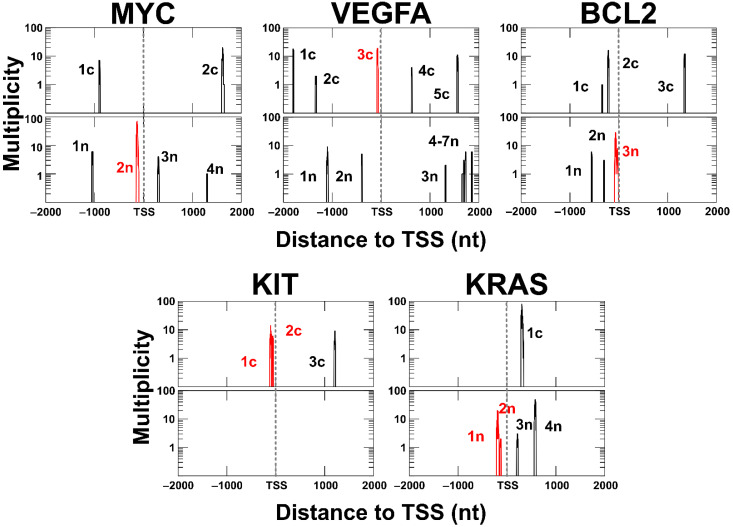
Multiplicities of G residues located within promoter regions of biologically relevant genes discussed in this paper plotted as a function of position relative to the transcription start site (TSS). The coding strand is shown as the top panel and all G4CRs are labeled with a lowercase c. The non-coding strand is the bottom panel, and all G4CRs are labeled with a lowercase n. G4CRs were numbered from left to right, and multiplicity is plotted on a logarithmic scale. The G4CR corresponding to the most discussed in the literature is shown in red for each gene.

**Figure 8 ijms-23-16020-f008:**
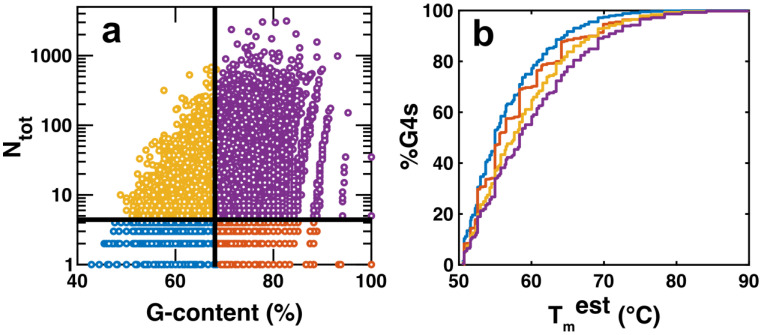
(**a**) Scatter plot of N_tot_ and G-content. N_tot_ is plotted on a logarithmic scale, whereas G-content is plotted on a linear scale. The graph is sectioned into four sections; G4CRs which contain less than the median value of N_tot_ and G-content (blue), G4CRs which contain more than the median value of N_tot_ but less than the median value of G-content (yellow), G4CRs which contain more than the median value of N_tot_ and of G-content (purple), G4CRs which contain less than the median value of N_tot_ but more than the median value of G-content (orange), (**b**) dependence of estimated melting temperatures for G4s contained in each of the different sections of (**a**). Colors represented in b are the same as those in (**a**).

**Table 1 ijms-23-16020-t001:** A subset of G4CRs with high N_tot_, N_tand_, G-content, and length that are found close to the transcription start site for genes which are dysregulated in a number of different cancers.

Gene	Strand	Length (nt)	G-Content (%)	N_tot_	N_tand_	Distance to TSS	Diseases
CAPN12	Non-coding	329	71	2363	13	−3	Pancreatic cancer [89]
USF2	Non-coding	90	80	907	4	−1	Small cell lung cancer, [90] breast cancer [91]
TTLL12	Coding	328	74	1311	14	3	Ovarian cancer [92]
ANO7	Coding	306	70	895	11	−48	Prostate cancer [93,94]
RAE1	Coding	127	83	3134	7	−290	Lymphoid and epithelial tumors [95]

## Data Availability

All data needed to evaluate the conclusions in the paper are present in the paper and/or the Appendix A. All code described in the methods section can be accessed at https://github.com/Christopher-Hennecker/GReg.

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
