# Peer review of "Structural Polymorphism of Guanine Quadruplex-Containing Regions in Human Promoters"

_ijms, 2022, doi:10.3390/ijms232416020_

Round 1

Reviewer 1 Report

By using a bioinformatics study the authors evaluated G4 structural polymorphism in promoter regions of the human genome. They defined G4 containing regions and calculated the degrees of G4 polymorphism depending on different parameters such as location, the length of loops and formation of bulges in the G4 structures.

The paper seems logically organized and the writing style is clear. The cited references are numerous. However, the formating of the current version needs to be improved.

In my opinion, the presented results are interesting for future bioinformatics studies, as well as for the broader G-quadruplex community. I therefore recommend the publication of this paper in the Special Issue "Bioinformatics of Unusual DNA and RNA Structures" of the journal IJMS.

Reviewer 2 Report

What is the role of pH on the guanine quadruplex-containing regions?

Reviewer 3 Report

The manuscript entitled "Structural Polymorphism of Guanine Quadruplex-Containing Regions in Human Promoters" presents a tool for predicting G-quadruplex structural polymorphism based on primary genomic sequences in promoter regions. G-quadruplex DNA is the most studied non-B-DNA motif because of its structural peculiarities observed in vitro, its apparent key role in a variety of physiological processes in vivo, and because of its therapeutic potential. The GReg algorithm presented in this manuscript helps G-quadruplex biologists to identify dynamic sequence regions in genomes that might correspond to the G-quadruplex motif. The work has some merit to be considered for publication. However, I have some suggestions for improving the manuscript.

Introduction: 

1.    G-quadruplexes are also observed in the origins of replication in higher eukaryotes. Review and cite relevant literature.

2.    Provide more information on   G-register,  exchange, and “spare tire” G-tracts. 

3.    Authors should review more recent tools presented in https://doi.org/10.1093/nar/gkz1097 in the introduction.

Results:

1.    The positional dependence of G4CRs for promoter regions of other model organisms with sufficient promoter datasets namely  Mus musculus, Drosophila melanogaster, Danio rerio, C elegans, A thaliana, Zea mays, Saccharomyces cerevisiae, and Plasmodium falciparum should be performed. This analysis reinforces their finding in other eukaryotes and makes this manuscript more comprehensive. How about comparing these profiles with shuffled promoter sequences?

2.    These authors focused on sequence-dependent dynamic G-quadruplex structural polymorphism in the promoter regions. It should be noted that G-quadruplexes can modulate gene expression in both positive and negative ways. From their results, it seems that both coding and non-coding strand have similar signatures. Justification required. 

3.   In GReg algorithm webser implementation (http://172.105.0.21/input/), provide default values of loop size, bulge, and  temp for non-G-quadruplex experts.

4.    The server ideally should accept Fasta formatted sequence also. Is there any size limitations to the input data?

5.    Provide more information on web server implementation.

6.  Discuss more anticipated applications of the tool in the discussion section.

Round 2

Reviewer 3 Report

Authors have incorporated all the concerns raised by this reviewer.   I congratulate them for this excellent work.